# Chromatin accessibility established by Pou5f3, Sox19b and Nanog primes genes for activity during zebrafish genome activation

Máté Pálfy[1‡], Gunnar Schulze[2‡], Eivind Valen[2,3], Nadine L. Vastenhouw[1]*

**1** Max Planck Institute of Molecular Cell Biology and Genetics, Dresden, Germany, **2** Computational Biology Unit, Department of Informatics, University of Bergen, Bergen, Norway, **3** Sars International Centre for Marine Molecular Biology, University of Bergen, Bergen, Norway

‡ These authors share first authorship on this work.
* vastenhouw@mpi-cbg.de

**Data Availability Statement:** All raw and processed sequencing data generated in this study have been submitted to the NCBI Gene Expression

## Abstract

In many organisms, early embryonic development is driven by maternally provided factors until the controlled onset of transcription during zygotic genome activation. The regulation of chromatin accessibility and its relationship to gene activity during this transition remain poorly understood. Here, we generated chromatin accessibility maps with ATAC-seq from genome activation until the onset of lineage specification. During this period, chromatin accessibility increases at regulatory elements. This increase is independent of RNA polymerase II-mediated transcription, with the exception of the hypertranscribed miR-430 locus. Instead, accessibility often precedes the transcription of associated genes. Loss of the maternal transcription factors Pou5f3, Sox19b, and Nanog, which are known to be required for zebrafish genome activation, results in decreased accessibility at regulatory elements. Importantly, the accessibility of regulatory regions, especially when established by Pou5f3, Sox19b and Nanog, is predictive for future transcription. Our results show that the maternally provided transcription factors Pou5f3, Sox19b, and Nanog open up chromatin and prime genes for activity during zygotic genome activation in zebrafish.

## Author summary

In eukaryotes, DNA is packed inside the cell nucleus in the form of chromatin, which consists of DNA, proteins such as histones, and RNA. Chromatin accessibility influences when and where DNA-binding proteins such as transcription factors and RNA polymerase II find their targets in order to activate transcription. It is unclear, however, whether the accessibility of regulatory regions precedes and predicts future transcription. To address these questions, we took advantage of ATAC-seq, which is a powerful technique to probe chromatin accessibility states. We analyzed zebrafish embryos as they gear up to start transcription for the first time and related the accessibility of regulatory regions to the activity of associated genes. This revealed that chromatin at regulatory regions is often accessible prior to gene activity and that this has predictive value for future transcription.

Omnibus (GEO; http://www.ncbi.nlm.nih.gov/geo/)
under accession number GSE130944.

**Funding:** This work was supported by the Max
Planck Society (https://www.mpg.de/en), a Human
Frontier Science Program Career Development
Award to NLV, CDA00060/2012 (https://www.hfsp.
org/), the Bergen Research Foundation to EV
(https://www.mohnfoundation.no/?lang=en) and
the Norwegian Research Council to EV, #250049
(https://www.forskningsradet.no/en/). The funders
had no role in study design, data collection and
analysis, decision to publish, or preparation of the
manuscript.

**Competing interests:** The authors have declared
that no competing interests exist.

Analyzing transcription factor mutants, we further determined that Pou5f3, Sox19b and
Nanog play an important role in opening up chromatin. Thus, our study shows that tran-
scription factors open regulatory elements to prime gene activity during development.

## Introduction

Chromatin structure is an important determinant of eukaryotic transcription regulation. At
gene regulatory regions such as promoters and enhancers, chromatin needs to be in an accessi-
ble state to allow for binding of the transcriptional machinery. Consequently, the vast majority
of transcription factor (TF) binding co-localizes with accessible chromatin regions [1,2]. Chro-
matin accessibility is regulated through the interplay of modifications to nucleosomes, such as
posttranslational histone modifications and the incorporation of histone variants, as well as
the binding of trans-acting factors [3].

While a number of studies suggest that chromatin accessibility generally correlates with
gene activity [4–6], others have suggested that this correlation is, at least globally, rather weak.
For example, chromatin accessibility landscapes are broadly similar in cell types and tissues
with different expression signatures, and in embryonic cells exposed to different transcription
factor concentrations [2,7,8]. This argues against the use of chromatin accessibility as a proxy
for gene activity. In fact, open chromatin can also be detected at regulatory elements that
become active only later during development. During hematopoiesis, for example, accessibility
is more highly correlated with the 'poised' enhancer mark H3K4me1 than with the active
H3K27ac mark [9]. Furthermore, *Drosophila*, mouse, and human embryos harbor accessible
chromatin at regulatory elements prior to gene activation [4,10–13]. This raises the question
whether chromatin accessibility can prime genes for activation.

The onset of transcription during zygotic genome activation (ZGA) provides an excellent
system to address this question. In zebrafish, the bulk of zygotic transcription is initiated
around three hours post fertilization (hpf) at the 1000-cell stage [14]. However, zygotic genome
activation is a gradual process with different genes starting to be transcribed at different times
[15–18]. It starts as early as the 64-cell stage, when the miR-430 gene cluster is activated [19–
22], and ends when the cells in the embryo start to adopt different fates during gastrulation.
Recently, Pou5f3, SoxB1 and Nanog have been identified as key transcription factors involved
in activation of the zygotic genome; they bind thousands of putative regulatory elements, and
loss of these TFs results in the reduced expression of many genes [15,23–25]. Furthermore,
zebrafish Pou5f3 and Nanog have been shown to play a role in promoting an open chromatin
state at regulatory elements [26,27]. It remains unclear, however, whether Pou5f3, Sox19b and
Nanog prime genes for activity.

Here we show that chromatin accessibility at promoters and enhancers often precedes tran-
scriptional activity during zebrafish ZGA. The establishment of accessible regions requires the
transcription factors Pou5f3, Sox19b and Nanog, and primes genes for future activation.

## Results

### Chromatin accessibility at gene regulatory regions increases during genome activation

To study the dynamics of the chromatin landscape during zebrafish zygotic genome activation
(ZGA), we generated genome-wide chromatin accessibility maps by ATAC-seq [28]. Data
from bulk ATAC-seq reflects the average accessibility of cells in a population [3]. We analyzed

seven stages from before the onset of transcription (256-cell), during genome activation (high, oblong, sphere, dome), until the cells in the embryo start to adopt different fates during gastrulation (shield, 80% epiboly) (Fig 1A). Biological replicates showed a large degree of correlation and samples from adjacent stages clustered together (S1A and S1B Fig), demonstrating the high quality of our data. We called ATAC-seq peaks by MACS2 [29], using genomic DNA as an input control (S2 Fig and Methods) and used k-means to cluster these peaks. We identified five clusters with different chromatin accessibility dynamics (Fig 1B). The genomic regions in cluster 5 are only moderately accessible throughout the time series, while the accessibility of genomic regions in clusters 1 through 4 increases with different temporal dynamics. The regions in cluster 3 and 4 become accessible at different times during gastrulation. Cluster 2 regions become accessible during ZGA, while regions in cluster 1 are already moderately accessible before ZGA (at 256-cell stage) and increase rapidly in accessibility in subsequent stages. Focusing on regulatory elements (S3 Fig), we found that promoters (defined as +/- 1kb of the TSS) and putative enhancers (defined based on accessibility and binding of developmental TFs, see Methods) open with different temporal dynamics in clusters 1 through 4 (Fig 1C and 1D). This dynamic opening of regulatory elements is also apparent at individual gene loci (Fig 1E). We conclude that during genome activation, chromatin accessibility dynamically increases at gene regulatory regions.

## RNA polymerase II activity is generally not required for chromatin accessibility

The increase in chromatin accessibility that we observe during genome activation suggests that accessibility could be a consequence of transcriptional activity. To test this possibility, we inhibited transcription by injecting embryos with α-amanitin at the one-cell stage. α-amanitin treatment resulted in transcription inhibition and developmental arrest at sphere stage (S4 Fig), as previously described [15,30,31]. We collected embryos at oblong stage, performed ATAC-seq, and compared chromatin accessibility in α-amanitin-treated and untreated embryos. This revealed that in general, chromatin accessibility at oblong stage is not a consequence of transcriptional activity (Fig 2A). Of all regions that are accessible in untreated embryos at oblong stage, only the miR-430 cluster shows a dramatic loss in accessibility upon transcription inhibition (Fig 2A). Interestingly, the expression level of miR-430 is orders of magnitude higher than that of any other gene during early zebrafish development [19]. This suggests that the transcription-dependent chromatin accessibility at the miR-430 cluster could be the result of RNA polymerase II complexes that evict nucleosomes, as has been demonstrated *in vitro* [32–34], and is reminiscent of the transcription-dependent accessibility at highly transcribed MERVL retrotransposons at the onset of ZGA in mouse embryos [4]. To determine whether transcribing RNA polymerase II contributes to chromatin accessibility at other highly transcribed genes as well, we grouped genes based on their expression level at oblong stage and associated putative enhancers with these genes using the nearest neighbor approach (see Methods). This revealed that chromatin accessibility at promoters and enhancers is not affected by transcription inhibition, with the exception of the miR-430 locus (Fig 2B). This is exemplified in genome browser snapshots of chromatin accessibility at the miR-430 locus (hypertranscribed), *id1* (strongly expressed), and *fbxw4* (weakly expressed) loci in transcription-inhibited and untreated embryos (Fig 2C). We conclude that, apart from the miR-430 cluster, the chromatin accessibility landscape that is present during zebrafish genome activation does not depend on transcribing RNA polymerase II.

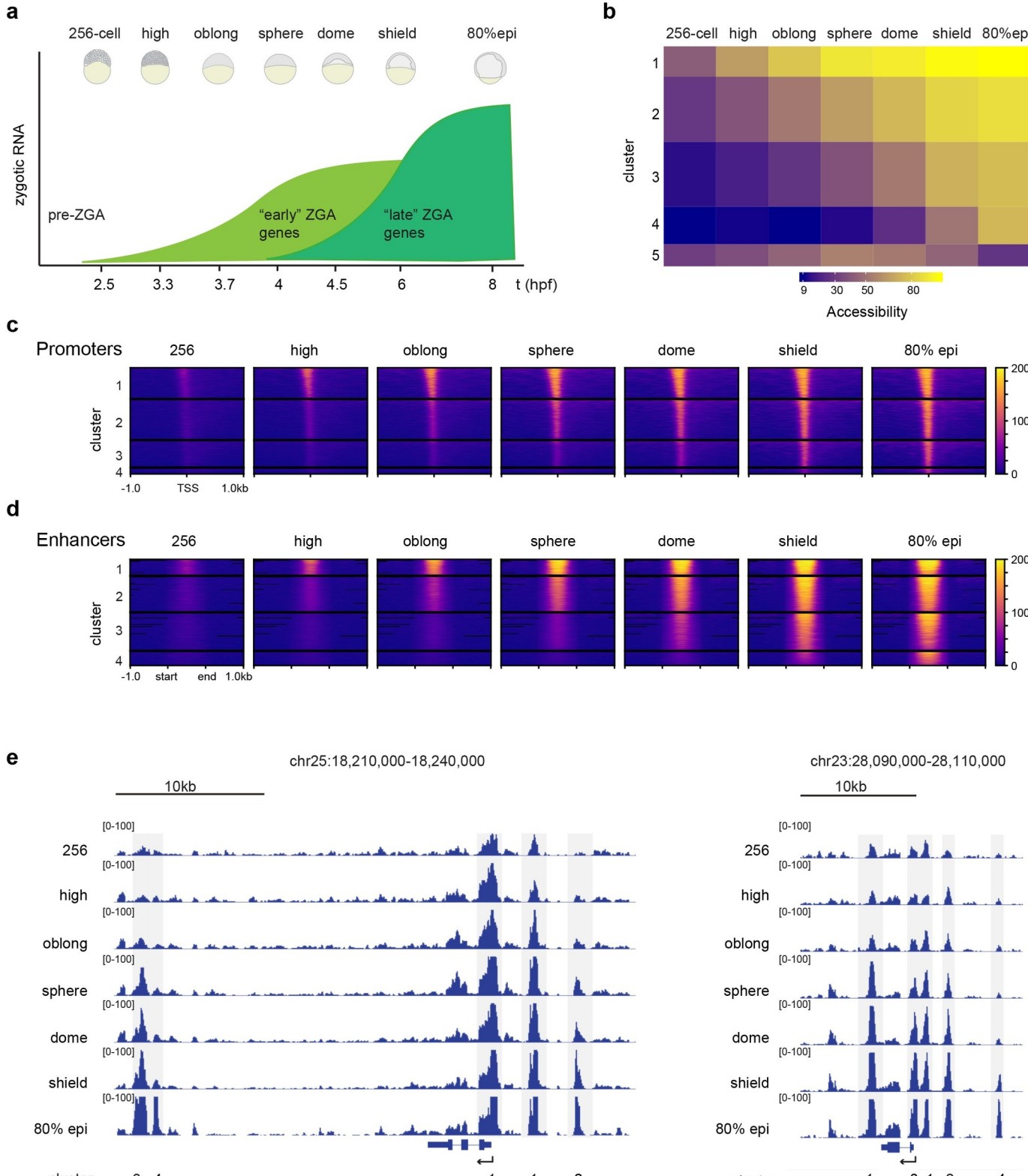

**Fig 1. Chromatin accessibility at regulatory elements increases during zygotic genome activation. a)** Schematic of timepoints and embryonic stages chosen for ATAC-seq. "Early" ZGA genes correspond to genes activated until sphere stage and "late" ZGA genes correspond to genes activated after sphere stage. **b)** Five clusters of chromatin accessibility over the ATAC-seq time-series. **c)** TSS-centered heatmaps of ATAC-seq peaks in clusters 1 through 4. **d)** Heatmaps of ATAC-

seq peaks centered on putative enhancer regions in clusters 1 through 4. **e)** Genome browser snapshots of two genomic regions over the ATAC-seq time-series. Peaks belonging to clusters 1–4 are indicated.

## Chromatin accessibility at regulatory elements precedes transcriptional activity

The observation that the increase in chromatin accessibility at promoters and enhancers is independent of transcriptional activity prompted us to investigate the temporal relationship between the opening of chromatin at regulatory elements and transcription onset. We selected genes that become transcriptionally active between sphere and shield stage [15] ("late" ZGA genes, see Fig 1A) and related chromatin accessibility at promoters with the time of gene activation. We found that promoters are accessible several stages prior to the activation of transcription (Fig 3A). Moreover, chromatin accessibility increases during the stages preceding transcription activation (Fig 3A). As a control, we selected promoters of genes that are not expressed until two days post fertilization [16]. At these promoters, chromatin accessibility is extremely low and does not change from 256-cell to 80% epiboly. Thus, the changes in accessibility at promoters are not due to a general increase in accessibility. Next, we used the nearest neighbour approach to identify the enhancers associated with the late ZGA genes and compared their accessibility with the time of gene activation. We found that enhancers, like promoters, are accessible prior to gene activation (Fig 3B). As for promoters, accessibility of

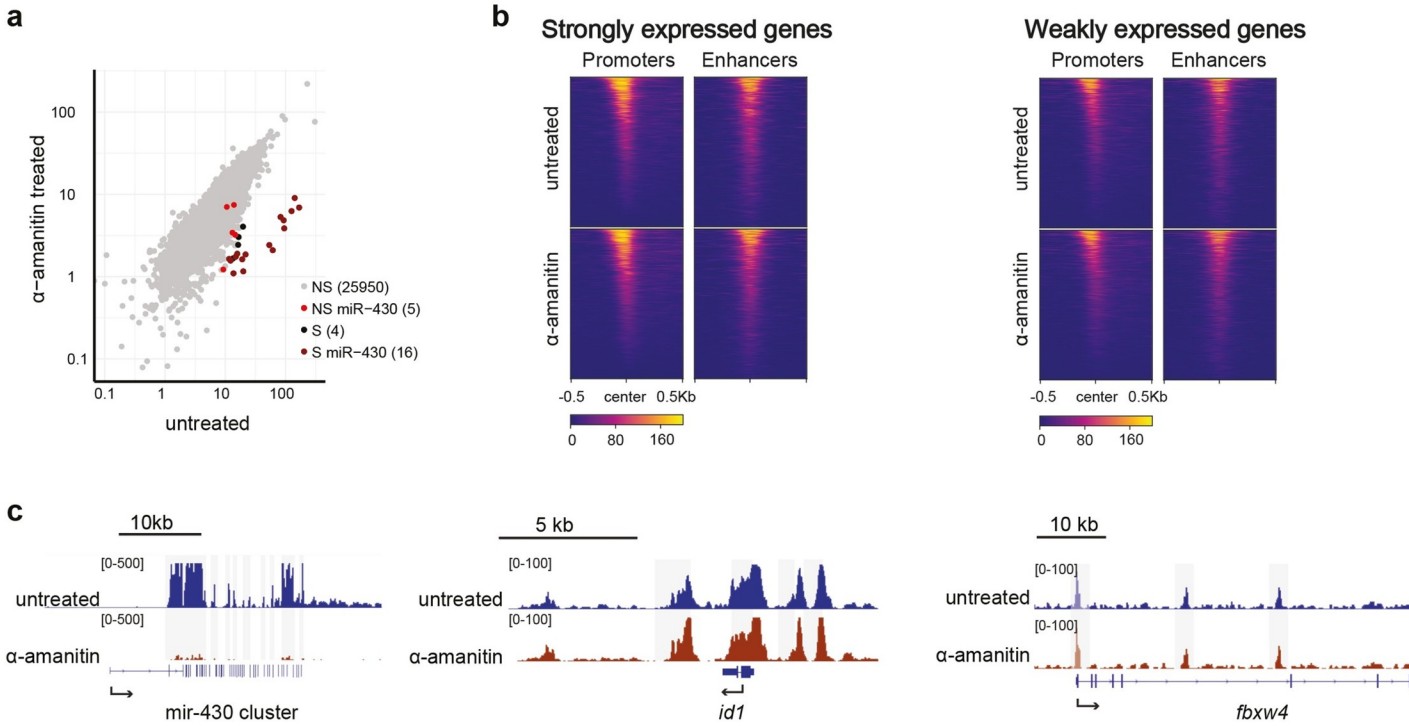

**Fig 2. Transcription inhibition affects chromatin accessibility only at the miR-430 locus. a)** Correlation dot plot comparing accessibility in α-amanitin-treated embryos with untreated embryos. Significantly (S) and non-significantly (NS) affected peaks (DESeq2 log2 fold change -1.5, 5% FDR), as well as peaks that fall into the miR-430 cluster are indicated. **b)** Heatmaps of chromatin accessibility in untreated and α-amanitin-treated embryos, grouped based on the expression level of associated genes at sphere stage (expression data taken from [15]; strongly expressed genes are defined as the top 20% genes expressed at sphere stage (RPKM > = 0.419), whereas weakly expressed genes are all other genes (with a minimal expression of RPKM > = 0.004 at sphere stage). **c)** Genome browser snapshots of untreated and α-amanitin-treated embryos at oblong stage for the hypertranscribed miR-430 gene cluster; *id1*, a gene with strong zygotic expression; and *fbxw4*, a gene with weak zygotic expression.

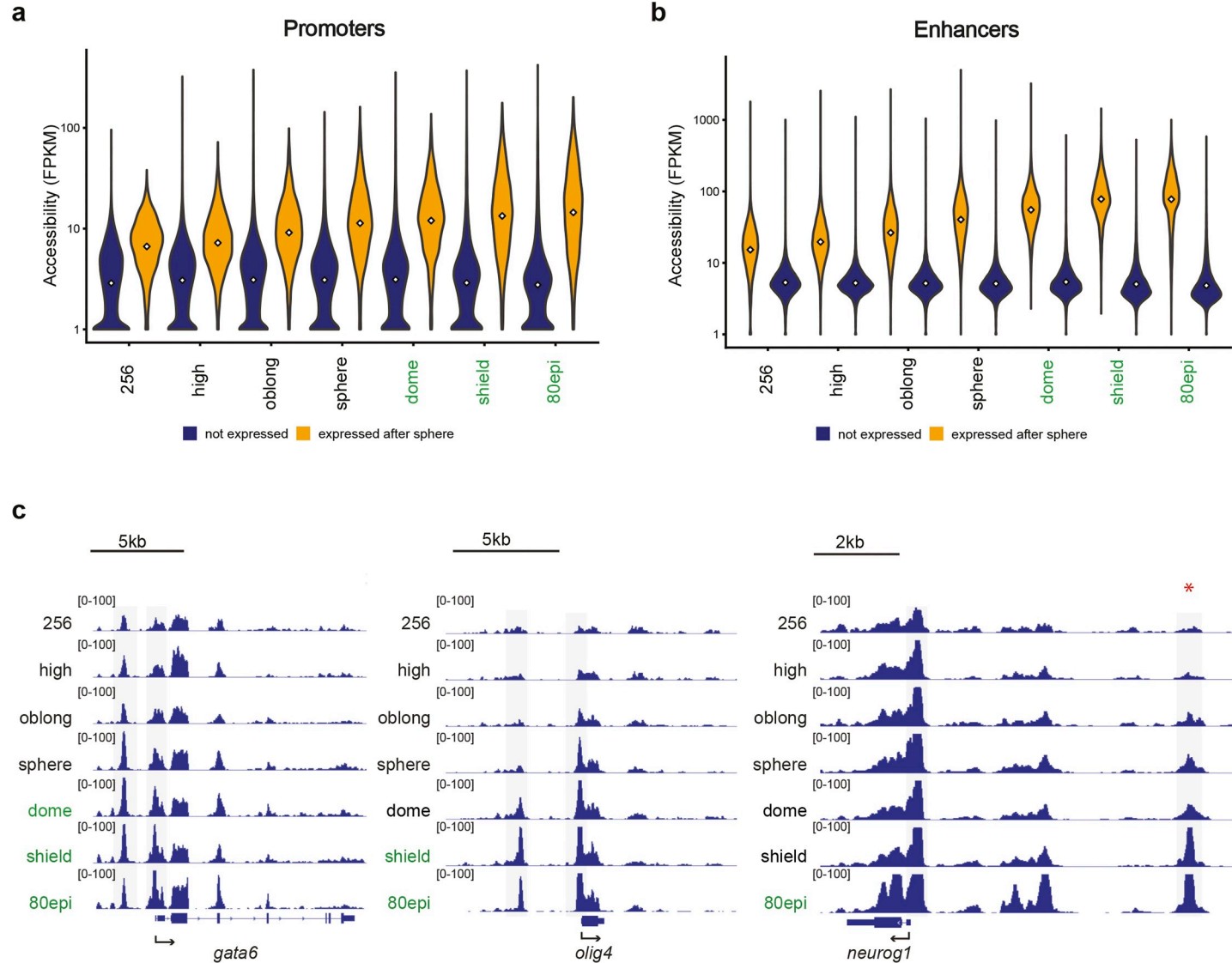

**Fig 3. Chromatin accessibility at regulatory elements increases prior to gene activity. a)** Distribution of accessibility values (ATAC-seq FPKM) of promoters associated with genes that become activated after sphere stage (yellow), and genes that are not expressed (blue). White squares represent the median value. **b)** Distribution of accessibility values of enhancers associated with genes that become activated after sphere stage (yellow), and genes that are not expressed during early development (blue). White squares represent the median value. **c)** Genome browser snapshots of chromatin accessibility at the *gata6*, *olig4* and *neurog1* gene loci; stages at which genes are transcribed are indicated with green fonts, the red asterisk marks an experimentally validated *neurog1* enhancer [37].

putative enhancers increases during the stages prior to gene activation. Here, we used zebrafish enhancers from adult tissues as a control [35]. At these enhancers, chromatin accessibility did not increase from 256-cell to 80% epiboly, confirming that the changes in accessibility at enhancers are not due to a general increase in accessibility. The accessibility of promoters and enhancers prior to gene activation, as well as the increase in accessibility prior to transcription activation, are easily visible at specific gene loci. For example, the promoter and putative enhancer of *gata6*, a gene that is activated at dome stage [16,36], are accessible at 256-cell stage and their accessibility increases prior to transcription (Fig 3C). A similar relationship between accessibility and transcription activation can be observed for the regulatory elements of *olig4* (activated at shield stage), and experimentally validated enhancers of *neurog* (activated at 80%

epiboly stage), *ta* (activated at sphere stage), and *otx2* (activated at 80% epiboly stage) [37–39] (Fig 3C, S5 Fig). We conclude that promoters and enhancers are accessible prior to gene activation, and that their accessibility increases during the stages preceding transcription activation.

Our results so far raise two important questions. First, which factors are involved in opening up chromatin at promoters and enhancers? And second, does accessibility predict future transcription?

## Pou5f3, Sox19b, and Nanog regulate chromatin accessibility

To address the first question, we set out to investigate the role of Pou5f3, Sox19b and Nanog in opening up chromatin. These transcription factors are maternally loaded, and have been shown to be required for the activation of many zygotic genes [15,23,25]. Moreover, their motifs are enriched in the distal regions that increase in accessibility between 256-cell and oblong stage (S6 Fig). We created *pou5f3* and *sox19b* mutants using CRISPR-Cas9 [40] (S7 Fig). We excised the entire coding region of the *pou5f3* and *sox19b* genes, thereby preventing any potential genetic compensation that could be triggered by mutant mRNA [41,42], or the generation of unexpected transcripts that escape nonsense-mediated decay [43]. Maternal-zygotic (MZ) *pou5f3* embryos arrest in their development during late gastrulation, as previously described [44–46]. In contrast, MZ*sox19b* embryos develop normally until adulthood. The lack of a strong MZ*sox19b* phenotype can be explained by the zygotic expression of *sox19a*, *sox3* and *sox2*, which all belong to the SoxB1 family and can compensate for each other's function when expressed [47]. For our studies this redundancy does not cause a problem, because at the early stages of development only Sox19b is present at high levels [15,47]. For *nanog*, we obtained a recently published null mutant that harbors a 10 base pair deletion in the first exon, causing a frameshift [48]. Importantly, the *nanog* mutant phenotype has previously been shown to be identical to embryos in which Nanog is knocked down by translation-blocking morpholinos [25,48], suggesting that no mechanisms are involved in compensating for this mutation. Having obtained bonafide mutants for *pou5f3*, *sox19b* and *nanog*, we performed ATAC-seq on MZ*pou5f3*, MZ*sox19b* and MZ*nanog* embryos at oblong stage and compared chromatin accessibility in mutants and wild-type embryos. Oblong stage is the last embryonic stage during which development relies entirely on maternal products, which makes it unlikely that effects observed in the mutants would be indirect. Upon the loss of Pou5f3, we observed a widespread decrease in chromatin accessibility (Fig 4A, S8 Fig). Similarly, the loss of Sox19b and Nanog resulted in a strong decrease in accessibility at hundreds of genomic regions (Fig 4A, S8 Fig). When analyzing changes in accessibility at regulatory elements, we found that promoters and enhancers are strongly affected (Fig 4B), as is apparent at genomic loci at Pou5f3-, Sox19b- and Nanog-regulated genes (Fig 4C). As expected, these changes in accessibility are significantly larger at promoters and enhancers of genes that are regulated by the respective factors [15] (S9 Fig).

In a previous study, analysis of the effect of transcription factors on chromatin accessibility was limited to sites that are bound by Pou5f3, Sox19b and Nanog [27]. As a consequence, the effects on promoter accessibility were not addressed. To determine whether the effect of Pou5f3 on promoter accessibility that we observed (Fig 4C) could also be observed in the dataset that was previously generated [27], we reanalyzed the MNase-seq data generated in this study. This revealed that distal as well as proximal regions that decrease in accessibility in our MZ*pou5f3* mutant show increased nucleosome occupancy in the *pou5f3* mutant (MZ*spg*) at dome stage (S10 Fig). We conclude that Pou5f3, Sox19b, and Nanog are important for the opening of chromatin at regulatory elements during zygotic genome activation.

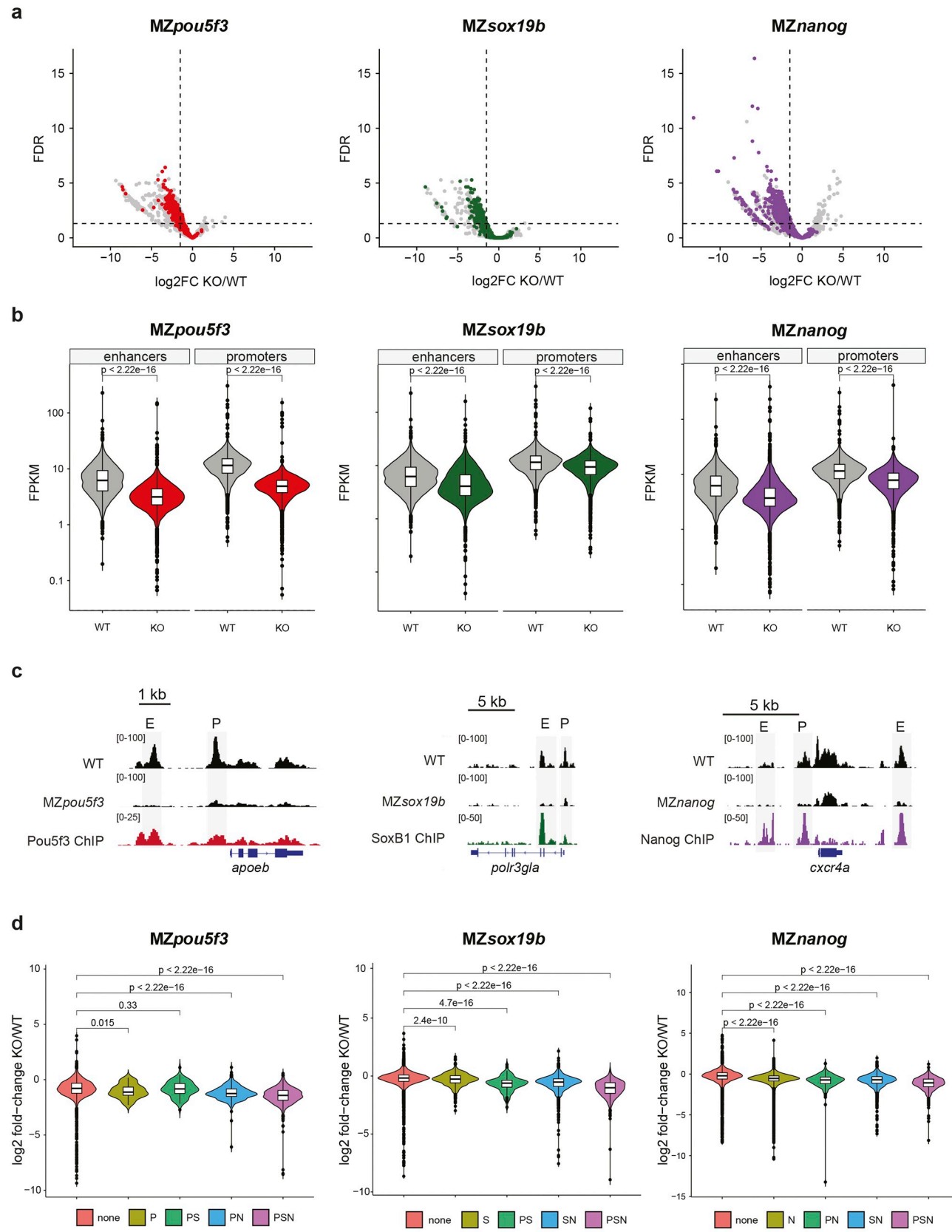

**Fig 4. Pou5f3, Sox19b and Nanog regulate chromatin accessibility at gene regulatory elements. a)** Volcano plots showing the log2 fold change in accessibility in MZ*pou5f3*, MZ*sox19b* and MZ*nanog* mutants compared to wild-type embryos at oblong stage. Sites bound by Pou5f1, SoxB1, and Nanog are indicated in red, green and purple respectively. Non-bound sites are indicated in grey. **b)** Violin plots show the accessibility value (FPKM) at promoters and putative enhancers in wild-type embryos compared to the respective TF mutants. Significance of differences was tested using paired t-tests. **c)** Genome browser snapshots show accessibility tracks in MZ*pou5f3*, MZ*sox19b* and MZ*nanog* mutants and wild-type embryos at oblong stage, for target genes of the respective TFs [15,25]. ChIP-seq tracks for the respective TFs at blastula stage are from data in [23,24]. **d)** Violin plots show the log2 fold change in mutants over wild-type at sites bound by different combinations of Pou5f3, SoxB1 and Nanog. p-values are shown for differences between binding sites as assessed by one-sided Wilcoxon tests.

It was recently suggested that Pou5f3 and Nanog affect chromatin accessibility only at sites that are bound by a combination of Pou5f3, SoxB1, and Nanog [27]. To determine whether the effect of Pou5f3, Sox19b, and Nanog is indeed limited to these so called PSN sites, we compared chromatin accessibility in the transcription factor mutants at sites bound by one, a combination of two, or all three transcription factors. Consistent with the previous study, accessibility is affected significantly at PSN sites (Fig 4D). In addition, however, sites bound by a single TF, or a combination of two of the TFs, also show a significant decrease in accessibility in the MZ*pou5f3*, MZ*sox19b*, and MZ*nanog* mutants. Together, these data suggest that Pou5f3, Sox19b, and Nanog regulate chromatin accessibility at promoters and enhancers that are bound by one or more of these factors.

Finally, we analyzed the motifs at sites where Pou5f3, Sox19b or Nanog increase accessibility. Pioneer factors often target partial motifs [49]. During human stem cell reprogramming for example, Oct4 and Sox2 target partial DNA motifs on nucleosomes [50]. Similarly, a fraction of nucleosomal FoxA binding occurs at non-optimal motifs which are degenerate in at least two base positions [51]. In contrast, our de novo motif analysis identified only minimal deviations from the canonical binding sites with no significantly enriched partial motifs (S11 Fig). In addition, we observe the largest change in accessibility at sites with canonical binding motifs (S12 Fig). These results are in agreement with another study in zebrafish [27]. Together, this suggests that in zebrafish, Pou5f3, Sox19b and Nanog regulate chromatin accessibility predominantly at sites with high motif strength.

## Chromatin accessibility established by Pou5f3, Sox19b and Nanog predicts future transcription

Next, we addressed the question whether an increase in accessibility has predictive value for future transcription. To that end, we selected promoter regions that show an increase in ATAC-seq signal between 256-cell and oblong stage. We binned these regions into 20% quintiles, and analyzed the median expression intensity for associated genes at sphere and shield stages. This analysis shows that larger increases in chromatin accessibility between 256-cell and oblong stage are correlated with higher expression levels at sphere stage (Fig 5A). To assess whether this correlation is also valid for distal elements, we linked ZGA genes with putative enhancers and observed that, as for promoters, the mean expression intensity is higher for genes that are linked to enhancers that show larger changes in accessibility (Fig 5B). Similar results were obtained at shield stage (S13A and S13B Fig). We conclude that an increase in chromatin accessibility at regulatory regions is a predictor of gene activity.

The observed role of Pou5f3, Sox19b and Nanog in regulating chromatin accessibility in early embryos suggests that these transcription factors might be involved in priming. To test this possibility, we extended our analysis of the relationship between accessibility and future gene expression (Fig 5A and 5B, S13A and S13B Fig) with the effect of transcription factor loss. To this end, we resolved gene expression levels at sphere stage by the fold change in chromatin accessibility at promoters and enhancers between 256-cell stage and oblong, as well as

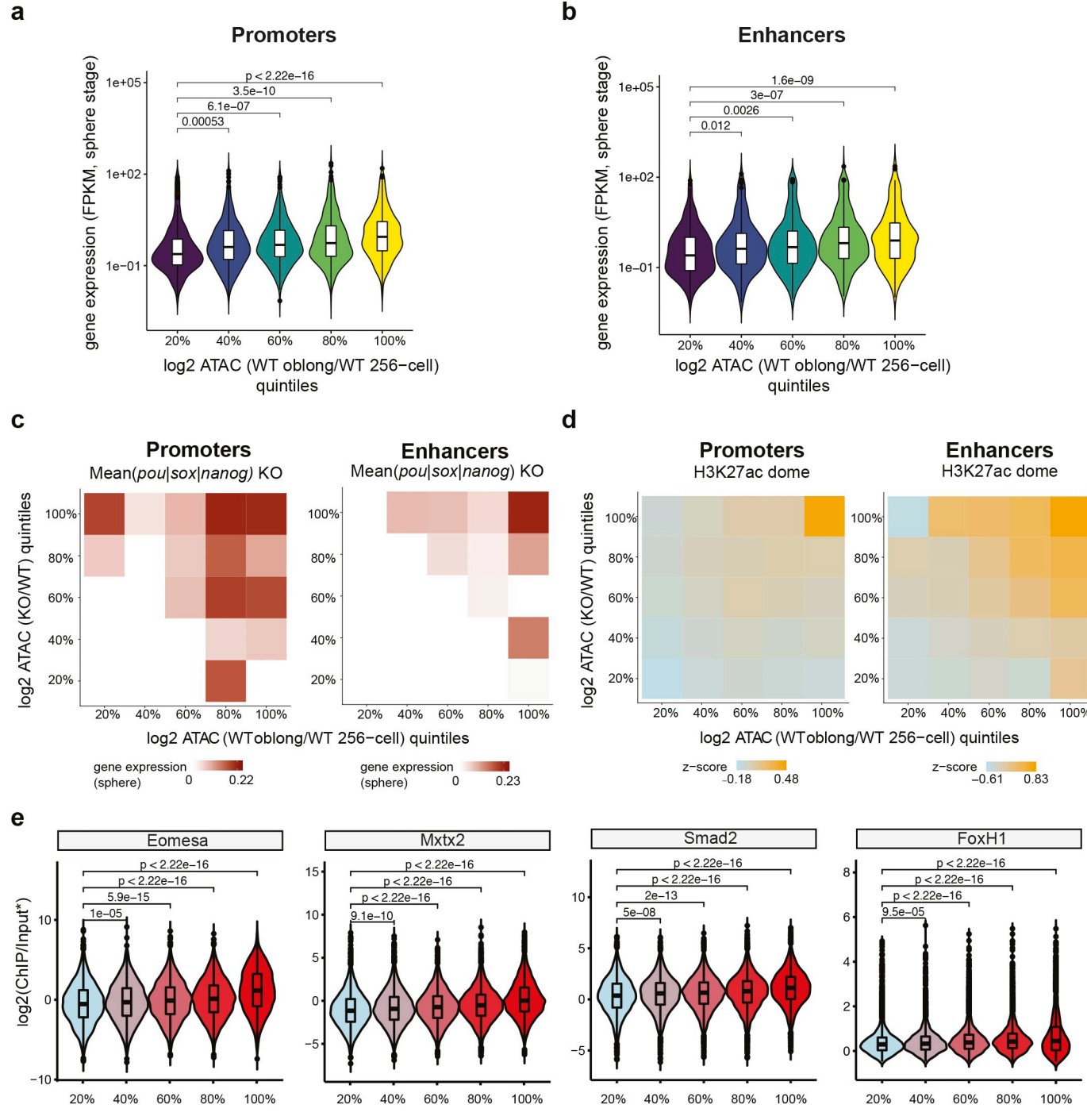

**Fig 5. An increase in chromatin accessibility predicts future gene expression, especially at Pou5f3/Sox19b/Nanog-regulated regions. a)** Promoter regions were sorted into 20% quintiles based on accessibility increase between 256-cell and oblong stage, and violin plots show the expression value of associated genes at sphere stage. p-values are shown for differences in expression between the quintiles as assessed by one-sided Wilcoxon tests. **b)** Putative enhancer regions were sorted into 20% quintiles based on accessibility increase between 256-cell and oblong stage and violin plots show the expression value of associated genes at sphere stage. p-values are shown for differences in expression between quintiles as assessed by one-sided Wilcoxon tests. **c)** Heatmaps show the median expression value for genes associated with regulatory regions at sphere stage. Genomic regions are resolved by 20% bins of accessibility increase between 256-cell and oblong stage (x-axis), and 20% bins of accessibility change in MZ*pou5f3*, MZ*sox19b* and MZ*nanog* mutants compared to wild-type embryos at oblong stage (y-axis). **d)** Heatmaps show the mean z-score normalized H3K27ac signal at the same genomic regions as in C. **e)** Regulatory elements were binned into 20% quintiles based

on accessibility change in MZ*pou5f3*, MZ*sox19b* and MZ*nanog* mutants compared to wild-type embryos, and violin plots show the enrichment (ChIP/Input) of Eomesa, Mxtx2, Smad2 and FoxH1 binding at blastula stages [24,81,82]. *FoxH1 signal was normalized to the genome-wide mean as no input sample was available. p-values are shown for differences in TF binding strength between the different bins as assessed by one-sided Wilcoxon tests.

by the average fold change in accessibility in Pou5f3, Sox19b, and Nanog mutants compared to wild-type embryos (Fig 5C). This revealed that regulatory elements that show both the largest increase in accessibility between 256-cell and oblong stage, and the largest reduction in accessibility upon the loss of either Pou5f3, Sox19b or Nanog, have the highest expression level at sphere stage (Fig 5C). Here too, analysis of shield stage embryos led to the same conclusion (S13C Fig). Because of the observed correlation between zygotic transcription and H3K27ac levels on regulatory elements during zebrafish ZGA [22,52], we repeated our analysis using H3K27Ac as a proxy for transcriptional activity. This independent analysis confirmed the important role of Pou5f3, Sox19b and Nanog in priming gene expression (Fig 5D). Together, this shows that Pou5f3, Sox19b and Nanog open up gene regulatory regions to facilitate gene activation.

Finally, if Pou5f3, Sox19b, and Nanog prime genes for activation it would be predicted that they enable the binding of other transcription factors. To test this hypothesis, we assessed whether regions where Pou5f3, Sox19b or Nanog open up chromatin are enriched for the binding of transcription factors involved in mesendoderm specification. This analysis showed that regulatory elements which experience the greatest loss in accessibility in MZ*pou5f3*, MZ*sox19b* or MZ*nanog* mutants are the most highly enriched for the binding of the transcription factors Mxtx2, Eomesa, Smad2 and Foxh1 at blastula stages (Fig 5E). Together, our results show that Pou5f3, Sox19b and Nanog prime genes for activation by opening up chromatin, which facilitates the binding of sequence-specific transcription factors.

## Discussion

In this study, we analyzed the dynamics and regulation of the chromatin accessibility landscape that underlies transcription activation during embryo development, using zebrafish ZGA as a model system. We find that promoters and enhancers increase in accessibility during genome activation. This is not due to transcriptional elongation by RNA polymerase II. In fact, accessibility at regulatory elements increases prior to gene activity. The increase in accessibility primes genes for activation, and Pou5f3, Sox19b and Nanog play a major role therein. Our results provide important insights into the relationship between chromatin accessibility and gene activity, and the molecular factors that shape the chromatin landscape during zebrafish genome activation.

### Zygotic genome activation is accompanied by a dynamic increase in chromatin accessibility at regulatory elements

Our finding that chromatin accessibility increases during genome activation, especially at promoter and enhancer elements, is consistent with recent studies in *Drosophila*, zebrafish, *Xenopus*, mouse and human embryos [4,10–13,26,53]. Because bulk ATAC-seq measurements reflect an ensemble average of distinct molecular states [3,54,55], an increase in accessibility over time may be due to an increase in the number of cell types in which specific regions are accessible, or a change to higher accessibility states of specific regions in most, if not all, cells of the embryo [3]. Individual zebrafish cells are transcriptionally largely uniform during early developmental stages [56,57], which suggests that the chromatin accessibility landscape may also be largely uniform, similar to what has been described for the early stages of *Drosophila*

development [58]. Moreover, the maternally loaded transcription factors Pou5f3, Sox19b and Nanog, which play an important role in opening chromatin, are uniformly expressed in the embryo at the time of ZGA [24,47,59]. Together, this favors a scenario in which the increase in chromatin accessibility during zebrafish genome activation is caused by an embryo-wide change in chromatin accessibility states.

### Transcribing RNA polymerase II evicts nucleosomes only at the hypertranscribed miR-430 locus

Our observation that chromatin accessibility is independent of transcriptional activity is consistent with studies that failed to detect a causal link between the transcription process and promoter chromatin accessibility at a genome-wide level [60,61]. However, *in vitro* and modeling studies have proposed that transcription by RNA polymerase II can form complexes which are able to evict nucleosomes and increase chromatin accessibility [32–34]. Our data suggest that such a mechanism may only take place at genomic regions that are very highly expressed. We find that the mir-430 cluster, which is transcribed two orders of magnitude higher than any other region, undergoes a dramatic loss in accessibility when elongation of RNA polymerase II is blocked. The miR-430 cluster is one of the first to be transcribed and initially, all transcribing RNA polymerase II localizes to the two alleles of the miR-430 cluster [20–22,62], which might facilitate the formation of RNA-pol II 'trains' capable of evicting nucleosomes, as shown *in vitro* [34].

### Chromatin accessibility precedes transcriptional activity

Our finding that regulatory elements can be accessible prior to transcription of associated genes is in agreement with the establishment of other "active" chromatin features prior to gene expression. At zebrafish promoters, for example, histone modifications H3K4me3 [63,64] and H3K27ac [62,65], a hypomethylated state [26,66–68], and well-positioned nucleosomes [69] can be established prior to the time their associated genes become expressed. Similar observations have been made in other species. In *Drosophila* embryos, for example, 47% of enhancers are accessible prior to large-scale genome activation [10], and lineage-specific chromatin regulatory landscapes are established prior to gastrulation [58]. Furthermore, histones can be acetylated prior to the major wave of ZGA at sites of early Zelda binding [70,71]. Similarly, a subset of promoters in human and mouse embryos exhibit open chromatin prior to activation of the genome [4,11,12]. We conclude that the establishment of accessible chromatin prior to gene activity is a common theme during early development.

### Pou5f3, Sox19b and Nanog establish chromatin accessibility

We found that zebrafish Pou5f3, Sox19b and Nanog are important for the increase in chromatin accessibility during ZGA. Ever since the discovery that human Oct4 and Sox2 act as pioneer factors during cell reprogramming [50,72], and the finding that their homologues are involved in zebrafish zygotic genome activation [15,23], it has been hypothesized that Pou5f3, Sox19b and Nanog could play a role in remodeling chromatin during genome activation. We now provide direct evidence for this hypothesis. Our results are in agreement with two recent studies reporting that the genetic loss of *pou5f3* and *nanog*, or their knockdown by translation-blocking morpholinos, influences the accessibility of chromatin at sites bound by these factors [26,27]. In contrast to the previously reported preference of Pou5f3 and Nanog to open up distal regions that are DNA methylated [26], we find that Pou5f3, Sox19b and Nanog are also important for regulating chromatin accessibility at promoters of zygotic genes, which are generally hypomethylated. This might explain why a third study [27] found no preference for

Pou5f3 and Nanog affecting accessibility at hypermethylated sites. Our finding that the loss of Pou5f3, Sox19b and Nanog has the largest effect on accessibility at sites bound by all three factors, is consistent with a previous study [27]. However, while this study reported that Pou5f3 and Nanog exclusively remodel chromatin accessibility at PSN sites, we find that they also regulate accessibility at regions that are bound by only one or two factors.

We note that our analysis of motifs that are enriched in regions that show an increase in chromatin accessibility during early ZGA suggests that in addition to Pou5f3, Sox19b, and Nanog, there are other transcription factors that could play a role in priming genes for activity. One such candidate factor is NFYA. It has previously been shown that NFYA binds to DNA in combination with TALE factors during zebrafish blastula stages [73,74], promotes chromatin accessibility and enhances the binding of Oct4, Sox2 and Nanog in embryonic stem cells [75], and establishes open chromatin during mouse ZGA [11]. Future studies will reveal whether NFYA cooperates with Pou5f3, Sox19b and Nanog in transcriptional priming during zebrafish ZGA.

### Pou5f3, Sox19b and Nanog prime genes for activation

We found that an increase in chromatin accessibility at regulatory elements primes genes for future activity. Although changes in chromatin accessibility have been correlated with transcription levels before [10,76,77], it was not clear whether chromatin accessibility at regulatory regions primes genes for activation. Here, we show directly that the increase in accessibility during the earliest stages of zygotic genome activation has a strong predictive value for future transcription. This is in contrast with the absolute level of chromatin accessibility prior to transcription, which does not show a correlation with transcription at later stages of zebrafish development [65,69].

Priming is strongest for genes where accessibility depends on Pou5f3, Sox19b and Nanog. Our data suggests that by opening up chromatin at regulatory elements, Pou5f3, Sox19b and Nanog facilitate the binding of other transcription factors involved in lineage specification and patterning. This is in agreement with recent work showing that Pou5f3 and Sox3 remodel chromatin during *Xenopus* ZGA to predefine competence for germ layer formation [53]. Future studies will be required to explore in more detail how Pou5f3, Sox19b and Nanog prime genes for activity. One potential mechanism is that these transcription factors mediate the establishment of H3K27ac at regulatory elements [62,65]. Pou5f3, Sox19b and Nanog, as well as H3K27ac are essential for zygotic genome activation [15,22,65], and the highest H3K27ac enrichment occurs at Pou5f3-, Sox19b-, and Nanog-primed loci (this study). The interdependence of these factors in the context of priming, however, remains unclear. Another potential mechanism involves the recruitment of chromatin remodelers to establish accessibility at regulatory elements. Recently, it was suggested that BRG1 could be involved in Pou5f3-mediated opening of chromatin in zebrafish [26], similar to what has been reported in ES cells [78]. However, it is likely that Pou5f3-mediated chromatin remodeling in zebrafish mechanistically differs from other species, given that zebrafish Pou5f3 lacks the linker region between the POU domains, which appears to be important for the interaction with Brg1 [79] and its capability for reprogramming [79,80].

## Materials and methods

### Ethics statement

All animal experiments were conducted according to the German Animal Welfare act and the European Communities Council Directive of 22 September 2010, and approved and licensed

by the local ethics committee (Landesdirektion Sachsen, Germany; license number DD24.1-5131/354/92).

## Zebrafish husbandry and manipulation

Zebrafish (TLAB) were maintained at 28.5°C according to standard protocols. Embryos were dechorionated with pronase and were grown in Danieau's medium on agarose dishes until the desired stage. For transcription inhibition, α-amanitin (A2263; Sigma) was injected at the 1-cell stage at a concentration of 0.2 ng per embryo. Staging of embryos was according to [83].

## Generation of MZ*pou5f3*, MZ*sox19b* and MZ*nanog* embryos

MZ*pou5f3* and MZ*sox19b* embryos were generated by CRISPR-Cas9. Two gRNAs were designed to target sequences close to the N-and C-terminal ends of both the *pou5f3* and *sox19b* genes using the CHOP-CHOP tool [84,85]. The following gRNA-s were used to delete the *sox19b* and *pou5f3* genes (the **NGG** PAM sequence indicated in bold):

N-terminal *sox19b*: GGGTTGCGTTGAGCGCTCAT**TGG**
C-terminal *sox19b*: GGTGTCCAACAGCACTATCT**TGG**
N-terminal *pou5f3*: GGCCATGTATCCTCAAGCCG**CGG**
C-terminal *pou5f3*: GGCTTGCACCCTGGTTTGGT**GGG**

gRNAs were cloned into the T7cas9gRNA2 vector [40] and in vitro transcribed using the MEGAshortscript T7 kit (Ambion/Invitrogen), followed by purification with the mirVana miRNA Isolation kit (Ambion/Invitrogen). Embryos at the early 1-cell stage were injected with 150 pg of Cas9 mRNA and 30 pg of each gRNA. Cas9 mRNA for the injections was produced by in vitro transcription of the p3T3S-nls-zCas9-nls plasmid [40] using the T3 mMessage machine transcription kit (Ambion/Invitrogen).

Embryos injected with Cas9 mRNA and gRNAs targeting *sox19b* and *pou5f3* were screened for mutant founders by outcrossing to wild-type fish. Maternal-zygotic mutants for *sox19b* were produced by incrossing homozygous mutant fish. Homozygous *pou5f3* mutant fish were rescued with injection of 40pg of Pou5f3-2xHA mRNA [31], and MZ*pou5f3* embryos were obtained by incrossing homozygous *pou5f3* mutant fish.

To obtain MZ*nanog* embryos, homozygous zygotic *nanog* mutants from [48] were incrossed.

## ATAC-seq

The ATAC-seq protocol was adapted from [86] and performed on approximately 70,000 cells. Zebrafish embryos were first deyolked using buffers as described in [87]; briefly, embryos were dissociated in deyolking buffer by brief vortexing and shaking at 1100 rpm at 4°C. Cells were collected by centrifugation at 400 rcf and the supernatant was removed. The cells were washed two more times with deyolking wash buffer. For the 256-cell stage, 300 embryos were manually deyolked as we were not able to obtain sufficient number of cells by buffer deyolking.

Following dissociation, cells were lysed in cell lysis buffer. Nuclei were spun down at 500 rcf for 10 minutes at 4°C, resuspended in tagmentation mix and incubated at 37°C for 30 minutes with gentle shaking at 800 rpm. Tagmented DNA was cleaned up using the Qiagen Minelute Reaction Cleanup kit and DNA was eluted in 10 ul of EB. DNA libraries were made by a limited cycle PCR Using the NEBNext High-fidelity 2X PCR Master Mix (NEB). To maximize the complexity of DNA libraries, the number of PCR cycles were determined by qPCR as described in [86]. Following 11–13 cycles of PCR, the libraries were purified using the Minelute Reaction cleanup kit (Qiagen) and measured with a fragment analyzer. Samples were sequenced following adaptor cleanup on an Illumina HiSeq25000 instrument with 75bp reads.

We performed two or three biological replicates for wild-type samples from different stages (S1 Fig), two biological replicates for amanitin-treated embryos (Pearson correlation = 0.95), two biological replicates for MZ*pou5f3* embryos (Pearson correlation = 0.96), two biological replicates for MZ*sox19b* embryos (Pearson correlation = 0.88), and two biological replicates for MZ*nanog* embryos (Pearson correlation = 0.96).

### RT-qPCR

25 embryos per stage were flash-frozen in liquid nitrogen. RNA was extracted using the RNeasy Mini kit from Qiagen. 1microgram of RNA was reverse transcribed to produce cDNA using the iScript cDNA Synthesis kit (Bio-Rad). For qPCR, a readily made SYBR green Mastermix with ROX (100nM) was used. Primers for *fam212aa*, *sox19a*, *mxtx2* and the housekeeping gene *eif4g2a* were from [31].

### ChIP-seq processing

We obtained ChIP-sequencing datasets for transcription factors from GEO studies GSE84619 (Ta, Tbx16 and Mixl1), GSE34683 (Nanog, Mxtx2), GSE51894 (Smad2, Eomesodermin), GSE67648 (Foxh1), GSE39780 (Pou5f1, Sox2). In addition we downloaded histone ChIP-seq for H3K27ac and H3K4me3 from GSE32483 for dome stage. Fastq-files were adapter and quality trimmed using cutadapt and aligned to the zebrafish genome build Zv10 using bowtie2 with parameters—no-mixed—no-discordant (paired-end datasets only). We deduplicated and filtered alignments by quality (q > = 30) using samtools. We called peaks using MACS2 at 5% FDR using ChIP-Input libraries as background (-c flag), if available. For histone ChIP-seq, we called peaks using the—broad flag.

### ATAC-seq processing

ATAC-seq paired-end reads were adapter- and quality trimmed using cutadapt and aligned to zebrafish genome build Zv10 using bowtie2 with a maximum insert size of 2000 (-X 2000) and parameters '—no-mixed—no-discordant'. Alignments were further deduplicated using 'samtools rmdup'. Only properly paired alignments with alignment quality > = 30 were used in downstream analyses. Additionally, alignments to the mitochondrial genome were filtered out.

Alignment files were converted to BED format using the 'bedtools bamtobed' utility in combination with a custom awk script. Start positions of reads mapping to the plus strand were adjusted by +4bp and end positions of reads mapping to the minus strand were adjusted -5bp to account for the transposase cutting event. Adjusted alignments were further subdivided into nucleosomal and open chromatin fragments corresponding to nucleosome-free regions (NFR) by taking a fragment size cut-off of 130bp (determined by visual inspection of the fragment length distributions).

### ATAC-seq peak calling

ATAC peaks were called for both all fragments and the NFR fraction only with the program MACS2 at an FDR of 5% with additional parameters "—no-model—no-lambda" and using the control digestion of genomic DNA as background (-c parameter). For each stage/condition we called peaks based on pooled replicates and subsequently overlapped the resulting peaks with those called in individual replicates, retaining only peaks that were confirmed in all replicates. We then merged all NFR peaks called in wild-type conditions into a consensus set (min

1bp overlap between peaks). We called a peak from the consensus set "present" in a particular stage if it overlapped by a replicated peak identified in that stage.

## Differential accessibility analysis

For differential analysis, we counted ATAC-seq reads over the consensus peak set using the featureCounts function from the Rsubread package. To enrich for fragments associated with nucleosome free regions, only fragments up to a length of 130bp length were counted. Alignments from multimapping reads were additionally discarded for this analysis. Differential analysis was performed using DEseq2, setting the size factor for each replicate to the total fragment count in that replicate. We considered peaks with a log2 fold change of -1.5 or 1.5 at an FDR of 5% as significantly decreasing or increasing, respectively.

## Definition of cis-regulatory elements and putative enhancers

We extracted transcription start sites (TSS) for all genes based on Danio rerio Zv10 genome assembly and ensembl v85 gene build. Promoters were defined as 2kb windows centered on the TSS. We determined putative enhancer elements as distal (min 1kb distance from any TSS) accessible regions from the consensus peak set that overlap at least one ChIP-seq peak of either Sox2[23], Nanog[24] or Pou5f1[23] and one additional developmental transcription factor (Smad2 [81], FoxH1[82], Mxtx2[24], Eomesodermin[81], Ta, Tbx16 or Mixl1[88]).

## Clustering of accessible regions

We calculated the average open chromatin fragments (fragment length $< = 130bp$) per kilobase per million reads mapped (FPKM) for each consensus peak in each stage and clustered the matrix of accessibility values by k-means clustering ('kMeans' function in R). We set the number of clusters k to 5 and initialized 10 random starts with a maximum of 1000 iterations. This resulted in 60% of the total variance explained by differences between clusters. All other parameters were kept at their default. The resulting clustering was then visualized in a heatmap. For this purpose, the matrix values were further binned into 100 quantiles ranging from low (0) to high (100) accessibility.

## Accessibility heatmaps

For each ATAC-seq library, we generated coverage tracks in bigwig format using the 'deeptools bamCoverage' utility at a bin size of 20bp. We extended alignments to cover the full fragment width (-e) and normalized to the individual library sizes using RPKM normalization. To separately analyze the open chromatin fraction, we additionally generated tracks using only fragments smaller or equal to 130bp in length. We plotted the open chromatin coverage signal over promoters and enhancers and at transcription factor binding sites, using the 'deeptools plotHeatmap' utility. For promoters and transcription factors, we centered the heatmap on the TSSs and ChIP-seq peak summits, respectively. Enhancer regions were scaled to each represent a 1000bp window. We used the 'deeptools bigwigCompare' utility to generate log2 ratio tracks of MZpou5f3 ATAC-seq and MZspg MNase-seq (external) samples over their wild-type counterparts at oblong and dome stage, respectively.

## Motif analysis

Motif analysis was performed using the MEME suite v5.03 [89]. To include the zebrafish Nanog-like motif in our analyses, we used a 60bp window centered on the top 5000 peaks called from 3.5hpf Nanog ChIP-seq data [24] as input to MEME (-minw 6 -maxw 15 -nmotifs

10). We identified the consensus motif corresponding to the previously reported Nanog-like motif by visual inspection and added the corresponding position weight matrix to a local copy of the JASPAR 2018 core vertebrate collection of motifs [90]. This custom collection was then used in motif enrichment and comparative analyses.

Proximal (+/-1kb around TSS) and distal (enhancer and intergenic) ATAC-seq peaks were sorted by their relative increase in accessibility between 256-cell and oblong stage and the top 20% of peaks within each category were selected for motif analysis. Enrichment analysis against the modified JASPAR database (containing Nanog, see above) was performed using AME (MEME-suite) using non-standard parameters—*kmer 2—evalue-report-threshold 10*. A custom R script was used in connection with R/TFBStools to summarize enriched factors at the motif family level. For each TF-family we reported the minimal adjusted p-value as well as the maximal % of peaks in the input set that have at least 1 motif from this family. *De novo* motif analysis was performed using the MEME-CHIP utility with parameters *-order 1 -meme-p 4 -ccut 0 -meme-mod zoops -meme-minw 6 -meme-maxw 15 -meme-nmotifs 10*.

We selected the top *de novo* motif for each TF family detected by matching motifs to their closest known counterparts in the database (TOMTOM). A custom R-script was used in connection with the R/ggseqlogo package to extract and plot position weight matrices for these motifs.

## De novo motif analysis for differentially accessible TFBS

We centered a 200bp window on summits of transcription factor binding sites for Pou5f1 (post ZGA) andSox2 (post ZGA), counted ATAC-seq open chromatin fragments within these regions and called regions significantly affected in our MZ*pou5f3* and MZ*sox19b* mutants compared to wt condition (DESeq2, log2 fold-change $< = -1.5$, FDR $< 5\%$). We subsequently performed *de novo* motif analysis on sequences extracted from both significantly affected and unaffected (log2 fold-change $> = 0$) regions using DREME (MEME-suite) with non-default parameters: *-m 5 -mink 3 -maxk 8*. For all significant motifs, we additionally calculated centralized enrichment relative to the peak summit using CENTRIMO (MEME-suite) and selected motifs with an e-value of 0.01 or less as centrally enriched.

## Binned motif score analysis

We extracted canonical motifs for Pou5FB1 (MA0792.1), Sox2 (MA0143.3) and the Pou5f1--Sox2 double motif (MA0142.1) from the JASPAR 2018 database. We then performed low-stringency motif scanning using FIMO (MEME-suite) across 200bp windows, centered on TFBS summits (see previous section) using these motifs. We reported all motif hits at a p-value threshold of 0.05. Positions of significant motif matches were parsed into R and the highest scoring match for each sequence and motif was selected. For each motif, we divided the range of observed motif scores into 10 bins of approximately equal frequency (quantile binning) and plotted the distribution of associated accessibility fold-changes in mutant compared to wild-type conditions (violin plots).

## Gene expression sets and transcriptional priming

We obtained gene expression data from [15] and ([16,24] and sorted genes based on their zygotic expression strength in sphere and shield stage, respectively. We then plotted the distribution of normalized ATAC-seq signal over promoters and enhancers of sphere and shield activated genes. Genes with a zygotic expression RPKM$> = 1.0$ [15] were considered expressed in sphere or shield stage, respectively. Shield-expressed genes were additionally filtered to only contain genes not previously expressed in sphere. Enhancers were assigned to promoters by

proximity within a maximal distance of 20kb. We summed the expression level of genes reported in [16] until 8hpf and defined a cut-off of RPKM < 0.1 to select genes not expressed until shield stage as a control group. For enhancers we used late-activated enhancers [35] as a control.

We additionally compared the accessibility fold-changes in our maternal-zygotic mutants per gene, stratified by whether the same gene's gene expression was negatively affected in mutants as reported before [15]. We tested the statistical significance of this difference by two-sample Welch t-tests and visualized the fold-change distributions using boxplots.

## Supporting information

**S1 Fig. Correlation of chromatin accessibility between ATAC-seq samples in early zebrafish embryos. a)** Pearson correlation between individual ATAC-seq replicates. Two biological replicates were generated for 256-cell, sphere, shield and 80% epiboly stage, and three biological replicates for high, oblong and dome stage. **b)** Pearson correlation between pooled replicates of different stages.
(TIF)

**S2 Fig. Pipeline for analyzing chromatin accessibility.** Chromatin and naked genomic DNA from zebrafish embryos was tagmented using Tn5 transposase, followed by sequencing and alignment to the genome (bowtie2). Fragments of sub-nucleosomal size (nucleosome free regions, NFR) were selected by applying a 130bp cut-off after visual inspection of the fragment length profiles. Peaks were called using MACS2 using the signal obtained from the naked DNA sample as background. Peaks called in each wild-type stage were merged across the time-course to generate a consensus peak set for further analysis.
(TIF)

**S3 Fig. Dynamics of accessibility increase at regulatory elements. a)** Average accessibility increase across the time-series (from 256-cell to 80% epiboly) over 200bp bins genome-wide (black), at promoters (red), and putative enhancer elements (blue). **b)** The coefficient of variation of normalized accessibility across the time course for promoter and putative enhancer associated peaks. A higher value indicates more variation in accessibility across stages.
(TIF)

**S4 Fig. Transcription inhibition in zebrafish embryos. a)** Gene expression levels of the zygotically activated genes *fam212aa*, *sox19a* and *mxtx2* in untreated and α-amanitin-treated embryos. α-amanitin treatment results in transcription inhibition. **b)** Brightfield images of zebrafish embryos at 6hpf in untreated and α-amanitin-treated embryos. α-amanitin treatment results in embryonic arrest at sphere stage.
(TIF)

**S5 Fig. Additional examples for the observed increase in accessibility at regulatory elements prior to gene activity.** Genome browser snapshots showing the *ta* and *otx2* gene loci. Green fonts indicate the stages when the gene is active. Experimentally verified enhancers for these genes [38,39] are marked by a red asterisk.
(TIF)

**S6 Fig. Motif analysis for regions with increasing accessibility.** Enrichment of TF motifs from the JASPAR vertebrates database (with the custom Nanog motif from [24]) at proximal (within +/- 1kb of TSS) and distal (enhancer/intergenic) regions that show an increase in accessibility between 256-cell and oblong stage. Individual motifs were summarized at the motif-family level and the minimum adjusted p-value for each family is indicated by the circle

size in the plot. The color scale ranging from blue (low) to red (high) indicates the % of peaks with at least one motif from the respective motif-family.
(TIF)

**S7 Fig. CRISPR deletion of *pou5f3* and *sox19b* genes.** Genome browser snapshots showing ATAC accessibility tracks at *pou5f3* and *sox19b* in wild-type and knock-out embryos. The absence of reads over the gene body of *pou5f3* and *sox19b* in knock-outs confirms the deletion of the loci. Location of the gRNAs used for deleting the genes are indicated with arrows.
(TIF)

**S8 Fig. Global change in chromatin accessibility in MZ*pou5f3*, MZ*sox19b* and MZ*nanog* mutants.** Violin plots show the aggregated fold change in accessibility in MZ*pou5f3*, MZ*sox19b* and MZ*nanog* mutants compared to wild-type embryos at oblong stage. Significance of differences was tested using paired, one-sided, t-tests.
(TIF)

**S9 Fig. Chromatin accessibility in mutants is most affected at regulatory regions of Pou5f3-, Sox19b- and Nanog-regulated genes.** Boxplots show the log2 fold change in accessibility in mutants compared to wild-type embryos (oblong stage) at putative enhancers and promoters associated with genes that are downregulated upon Pou5f3, Sox19b and Nanog knock-down (red) and genes that are not downregulated (grey). Expression data is from [15]. Statistical significance of the differences was tested by two-sample Welch t-tests.
(TIF)

**S10 Fig. Comparison of chromatin accessibility changes in maternal-zygotic *pou5f3* mutants assayed by ATAC-seq and MNase-seq. a)** Metagene profiles and heatmaps showing the log2 fold change in accessibility in MZ*pou5f3* mutants compared to wild-type at oblong stage. **b)** Metagene profile and heatmaps for the same regions as in a), showing the log2 fold change in nucleosome occupancy in *pou5f3* (MZ*spg*) mutants compared to wild-type at dome stage, using data from [27]
(TIF)

**S11 Fig. De novo motif analysis at loci with reduced chromatin accessibility in mutants identifies canonical motifs as most enriched.** De novo motif analysis was performed for peaks that are **a)** bound by Pou5f3 and strongly reduced (blue) or unaffected (red) in MZ*pou5f3* mutants mutants; **b)** bound by SoxB1 and strongly reduced (blue) or unaffected (red) in MZ*sox19b* mutants; or **c)** bound by Nanog and strongly reduced (blue) or unaffected (red) in MZ*nanog* mutants. Sequence logos of significant (e-value < 0.01) motifs identified are shown along with their associated E-value.
(TIF)

**S12 Fig. Regulation of accessibility by Pou5f3 and Sox19b is associated with high motif strength for these factors.** To determine the correlation between loss of accessibility in mutants and motif strength, we used motifs from the JASPAR 2018 database to perform low-stringency motif scanning. Binned motif scores are visualized from low (left) to high (right). Violin plots show the distribution of associated fold-changes in accessibility between **a)** MZ*pou5f3* and wild-type for the Pou5f1-Sox2 double motif; **b)** MZ*sox19b* and wild-type for the Pou5f1-Sox2 double motif; **c)** MZ*pou5f3* and wild-type for the Pou5FB1 motif; and **d)** MZ*sox19b* and wild-type for the Sox2 motif.
(TIF)

**S13 Fig. An increase in chromatin accessibility predicts future gene expression, especially at Pou5f3/Sox19b/Nanog-regulated regions. a)** Promoter regions were sorted into 20% quintiles based on accessibility increase between 256-cell and oblong stage, and violin plots show the expression value of associated genes at shield stage. p-values are shown for differences in expression between quintiles as assessed by one-sided Wilcoxon tests. **b)** Putative enhancer regions were sorted into 20% quintiles based on accessibility increase between 256-cell and oblong stage, and violin plots show the expression value of associated genes at shield stage. p-values are shown for differences in expression between quintiles as assessed by one-sided Wilcoxon tests **c)** Heatmaps show the median expression value for genes associated with regulatory regions at shield stage. Genomic regions are resolved by 20% bins of accessibility increase between 256-cell and oblong stage (x-axis), and 20% bins of accessibility change in MZ*pou5f3*, MZ*sox19b* and MZ*nanog* mutants compared to wild-type embryos at oblong stage (y-axis). (TIF)

## Acknowledgments

We thank Daria Onichtchouk for kindly providing zebrafish *nanog -/-* mutant fish, Stefan Hans for advice on CRISPR experiments and reagents, members of the Vastenhouw and Valen labs for discussions, Carine Stapel for manuscript comments, three anonymous reviewers for constructive feedback, and the following facilities and services: Fish Facility (MPI-CBG) and the DRESDEN-concept Genome Center.

## Author Contributions

**Conceptualization:** Máté Pálfy, Nadine L. Vastenhouw.

**Data curation:** Máté Pálfy, Gunnar Schulze.

**Formal analysis:** Gunnar Schulze.

**Funding acquisition:** Eivind Valen, Nadine L. Vastenhouw.

**Investigation:** Máté Pálfy, Gunnar Schulze.

**Methodology:** Máté Pálfy, Gunnar Schulze.

**Project administration:** Eivind Valen, Nadine L. Vastenhouw.

**Resources:** Máté Pálfy, Gunnar Schulze, Eivind Valen, Nadine L. Vastenhouw.

**Software:** Gunnar Schulze.

**Supervision:** Eivind Valen, Nadine L. Vastenhouw.

**Validation:** Máté Pálfy, Gunnar Schulze.

**Visualization:** Máté Pálfy, Gunnar Schulze.

**Writing – original draft:** Máté Pálfy, Nadine L. Vastenhouw.

**Writing – review & editing:** Máté Pálfy, Gunnar Schulze, Eivind Valen, Nadine L. Vastenhouw.

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
