## [Decision Letter · Decision Letter 0]

14 Aug 2019

Dear Dr Vastenhouw,

Thank you very much for submitting your Research Article entitled 'Chromatin accessibility established by Pou5f3, Sox19b and Nanog primes genes for activity during zebrafish genome activation.' to PLOS Genetics. Your manuscript was fully evaluated at the editorial level and by independent peer reviewers. The reviewers appreciated the attention to an important problem, but raised some substantial concerns about the current manuscript. Based on the reviews, we will not be able to accept this version of the manuscript, but we would be willing to review again a much-revised version. We cannot, of course, promise publication at that time.

If you decide to revise the manuscript for further consideration at PLOS Genetics, please aim to resubmit within the next 60 days, unless it will take extra time to address the concerns of the reviewers, in which case we would appreciate an expected resubmission date by email to plosgenetics@plos.org.

[LINK]

We are sorry that we cannot be more positive about your manuscript at this stage. Please do not hesitate to contact us if you have any concerns or questions.

Yours sincerely,

A. Aziz Aboobaker

Associate Editor

PLOS Genetics

Wendy Bickmore

Section Editor: Epigenetics

PLOS Genetics

Two of the expert reviewers have some substantial questions regarding the current manuscript. On balance I agree, that given what we already know, you could consider further analyses with the current data set to support your conclusions. I do not think it is necessary within the context of this study to provide further experimental validation of putative enhancers unless you already have these data available.

Reviewer's Responses to Questions

**Comments to the Authors:**

Reviewer #1: Zygotic genome activation is important for early embryonic development and the mechanisms underpinning this process should be of interest to the readership of PloS Genetics. Here Palfy and colleagues investigate ZGA in zebrafish, specifically focussing on the interplay between chromatin accessibility and transcription. They performed ATACseq on a series of early stages pre and post ZGA and found that from genome activation to onset of lineage specification chromatin accessibility increased in regulatory regions and preceded the transcription of associated genes. Using CRISPR mutants they show that this was dependent on the maternally expressed TFs, Pou5f3, Sox19B and Nanog, whose association with accessible elements was indicative of future transcription. They conclude that these TFs are involved in priming genes for transcription.

A lot is known about the timing of zygotic genome activation and Pou5f3, Sox19B and Nanog have previously been identified as key TFs involved. It was shown that they bind putative regulatory elements and a recent paper (Veil et al., Genome Research 2019) used MNase-seq and reports that these factors promote the open chromatin state. This study also provided chromatin accessibility data from similar stages (pre ZGA to dome stage). Thus, the present manuscript complements these previous reports and extends previous findings.

The experiments and associated bioinformatics analyses are well done, however, the overall advance is somewhat incremental.

They first attempt to address whether transcription is required for chromatin accessibility and compare a-amanitin treated and untreated embryos at the oblong stage, this treatment leads to arrest at sphere stage. Accessibility changes significantly only at the highly transcribed miR430 locus and not at any other highly transcribed genes. Thus, they conclude that RNA polymerase II activity is not required for chromatin accessibility, with the exception of the mir-430 cluster. It is not clear why they chose the id1 and fbxw4 loci to show ATACseq tracks and maybe the authors can comment on this selection also in the text (they do comment in the figure legend).

Looking at the timeline of developmental stages and publicly available expression data the authors observe that chromatin accessibility precedes transcription. They look at promoters and putative enhancers of genes that are activated at sphere and shield stages. The enhancers are defined by distance from TSS and presence of particular TF binding sites, but none of these putative enhancers is confirmed experimentally and their function and dependency on particular TFs is not confirmed.

They investigate the roles of Pou5f3, Sox19B and Nanog by creating CRISPR mutants followed by ATACseq at the oblong stage. The effects on accessibility at promoters and putative enhancers are quite small based on the volcano plots shown in Fig 4b, particularly in the MZsox19b and MZnanog mutants. Although for the selected genes shown in Fig 4c the effects are very clear and these genes and associated enhancers would be good candidates for experimental validation.

Overall, their analysis confirms a previous publication which suggested that these TFs regulate chromatin accessibility. In addition, the authors suggest that sites bound by a single TF or two of the TFs have reduced accessibility in the mutants, but this is not very clear from the plots shown in Fig 4d.

Finally, the authors find a correlation between increased accessibility of chromatin prior to gene expression at a subsequent stage and they suggest that Pou5f3, Sox19B and Nanog are priming gene expression.

The analysis reveals interesting associations, however the manuscript would benefit from experimental validations of some of the putative enhancers identified, including the confirmation of TF binding sites as being functionally important.

Reviewer #2: This manuscript investigates the process of zygotic genome activation (ZGA) in zebrafish embryos to explore the relationship between transcription, chromatin accessibility and how this is regulated, including the regulatory proteins involved. These are interesting and important questions because we still do not fully understand how chromatin accessibility is regulated and how changes in accessibility are related to gene expression during development i.e. whether open chromatin is a reliable predictor the genes being expressed in cells or the future genes that will be expressed.

The authors carry out ATAC-Seq in replicates at seven different stages of zebrafish embryogenesis covering from the 256 cell stage - approximately when zygotic gene expression begins – to gastrulation. I was curious however as to why earlier stages were not assayed since the authors themselves state that some zygotic transcription is observed earlier at the 64-cell stage. Comparison of the ATAC-Seq profiles across these stages allowed the authors to visualise and compare changes in open chromatin and how this is associated with changes in gene expression. They found five clusters of genes with different chromatin accessibility dynamics representing increasing numbers of genes with promoters and enhancers that become more accessible as development proceeds. They then provide convincing evidence that this increasing accessibility is generally not dependent on transcription except for the very highly expressed miR-430. They then show that accessibility of promoters and enhancers preceeds the expression of the associated genes. This is an important result that, consistent with data from other organisms, strongly suggests while generally open chromatin is not a great predictor of which genes are expressed at a given stage or tissue, it can be predictive of future gene expression during development.

The authors then investigate the effect of loss of three candidate transcription factors on chromatin accessibility during ZGA: Pou5f3, Sox19b and Nanog. They found that loss of any of these factors results in a decrease in chromatin accessibility and that this effect is caused by a loss of binding of these factors individually and in combination at different loci. Analysis of gene expression evidences that these three transcription factors likely prime genes for expression by increasing chromatin accessibility.

Overall I think this is a very interesting and well written paper that reports convincing data to advance our understanding of the regulation of gene expression during development.

Reviewer #3: N/A

**Have all data underlying the figures and results presented in the manuscript been provided?**

Reviewer #1: Yes

Reviewer #2: Yes

Reviewer #3: Yes

PLOS authors have the option to publish the peer review history of their article (what does this mean?). If published, this will include your full peer review and any attached files.

Reviewer #1: No

Reviewer #2: No

Reviewer #3: No

---

## [Decision Letter · Decision Letter 1]

2 Dec 2019

Dear Dr Vastenhouw,

We are pleased to inform you that your manuscript entitled "Chromatin accessibility established by Pou5f3, Sox19b and Nanog primes genes for activity during zebrafish genome activation." has been editorially accepted for publication in PLOS Genetics. Congratulations!

Yours sincerely,

A. Aziz Aboobaker

Associate Editor

PLOS Genetics

Wendy Bickmore

Section Editor: Epigenetics

PLOS Genetics

Comments from the reviewers (if applicable):

Reviewer's Responses to Questions

**Comments to the Authors:**

Reviewer #1: The authors have responded in detail to the comments made by reviewers. This is a well written manuscript which makes an important contribution.

Reviewer #3: The authors have addressed the vast majority of my concerns and provided explanations for the others. I am therefore happy with the manuscript in its present form.

**Have all data underlying the figures and results presented in the manuscript been provided?**

Reviewer #1: Yes

Reviewer #3: Yes

PLOS authors have the option to publish the peer review history of their article (what does this mean?). If published, this will include your full peer review and any attached files.

Reviewer #1: No

Reviewer #3: No

**Data Deposition**

http://datadryad.org/submit?journalID=pgenetics&manu=PGENETICS-D-19-00995R1

**Press Queries**

---

## [Editor Report · Acceptance letter]

2 Jan 2020

PGENETICS-D-19-00995R1 

Chromatin accessibility established by Pou5f3, Sox19b and Nanog primes genes for activity during zebrafish genome activation. 

Dear Dr Vastenhouw, 

We are pleased to inform you that your manuscript entitled "Chromatin accessibility established by Pou5f3, Sox19b and Nanog primes genes for activity during zebrafish genome activation." has been formally accepted for publication in PLOS Genetics! Your manuscript is now with our production department and you will be notified of the publication date in due course.

With kind regards,

Kaitlin Butler

PLOS Genetics

On behalf of:
